# Metabolic Adjustment of High Intertidal Alga *Pelvetia canaliculata* to the Tidal Cycle Includes Oscillations of Soluble Carbohydrates, Phlorotannins, and Citric Acid Content

**DOI:** 10.3390/ijms241310626

**Published:** 2023-06-25

**Authors:** Renata Islamova, Nikolay Yanshin, Elizaveta Zamyatkina, Ekaterina Gulk, Ekaterina Zuy, Susan Billig, Claudia Birkemeyer, Elena Tarakhovskaya

**Affiliations:** 1Department of Plant Physiology and Biochemistry, Faculty of Biology, Saint Petersburg State University, Universitetskaya Nab., 7/9, 199034 Saint Petersburg, Russia; renata.tag.isl@gmail.com (R.I.);; 2Faculty of Chemistry and Mineralogy, Leipzig University, 04103 Leipzig, Germanybirkemeyer@chemie.uni-leipzig.de (C.B.); 3German Center for Integrative Biodiversity Research (iDiv) Halle-Leipzig-Jena, 04103 Leipzig, Germany; 4Vavilov Institute of General Genetics, Saint Petersburg Branch, Russian Academy of Science, Universitetskaya Nab., 7/9, 199034 Saint Petersburg, Russia

**Keywords:** *Pelvetia*, brown algae, desiccation, tidal cycle, phlorotannins, metabolomics, citric acid, salicylic acid, titratable acidity, CAM-photosynthesis

## Abstract

The brown alga *Pelvetia canaliculata* is one of the species successfully adapted to intertidal conditions. Inhabiting the high intertidal zone, *Pelvetia* spends most of its life exposed to air, where it is subjected to desiccation, light, and temperature stresses. However, the physiological and biochemical mechanisms allowing this alga to tolerate such extreme conditions are still largely unknown. The objective of our study is to compare the biochemical composition of *Pelvetia* during the different phases of the tidal cycle. To our knowledge, this study is the first attempt to draft a detailed biochemical network underneath the complex physiological processes, conferring the successful survival of this organism in the harsh conditions of the high intertidal zone of the polar seas. We considered the tide-induced changes in relative water content, stress markers, titratable acidity, pigment, and phlorotannin content, as well as the low molecular weight metabolite profiles (GC-MS-based approach) in *Pelvetia* thalli. Thallus desiccation was not accompanied by considerable increase in reactive oxygen species content. Metabolic adjustment of *P. canaliculata* to emersion included accumulation of soluble carbohydrates, various phenolic compounds, including intracellular phlorotannins, and fatty acids. Changes in titratable acidity accompanied by the oscillations of citric acid content imply that some processes related to the crassulacean acid metabolism (CAM) may be involved in *Pelvetia* adaptation to the tidal cycle.

## 1. Introduction

The intertidal zone is a part of a shoreline lying between the high water and low water marks and thus covered with water at high tide and exposed to the air at low tide. The relative duration of underwater and waterless periods varies for the different subzones (low, mid, or high intertidal), and also depends on the tidal amplitude, which is maximal during spring tides and minimal during neap tides. Typically, the intertidal zones of the seas host rich and diverse ecosystems, though such a specific habitat may be considered among the most stressful ones due to the recurring changes of environmental conditions. Thus, seaweeds inhabiting this zone are constantly exposed to fluctuations of humidity, light intensity, temperature, salinity, nutrient availability, etc. [1,2].

The brown alga *Pelvetia canaliculata* is one of the species that has successfully adapted to the intertidal conditions. This miniature fucoid is abundant on the shores of the Arctic and North Atlantic seas, growing as a narrow belt on the wave-exposed rocks of the high intertidal zone. *Pelvetia* thalli lack airbladders and look like dichotomously branching bushlets of 2–15 cm high, firmly attached to the substratum with a small basal disc. The mature thalli can bear multiple receptacles on the tips of the branchlets. The color of the *Pelvetia* thalli varies from olive-yellow (wet) to dark brown (dried) [3]. An interesting ecophysiological feature of *P. canaliculata* is its intimate association with the endophytic ascomycete fungus *Stigmidium ascophylli*. Living in the form of such association called mycophycobiosis gives *Pelvetia* some lichen features and is supposed to contribute to the high tolerance of this organism [4,5].

Inhabiting the high intertidal zone, *Pelvetia* systematically faces extreme differences in environmental conditions. This part of the shore is covered with seawater for no more than 8 h per day during the spring tides and may remain dry for several days during the periods of the neap tides, thus *Pelvetia* spends most of its life exposed to air, where beyond desiccation, it is also subjected to high light and temperature stress in the daytime and sharp drops of temperature in the night. Adaptive responses of *P. canaliculata* to such a complicated environment are expected to require systematic rearrangements of all key physiological processes according to the phase of the tidal cycle. However, the physiological and biochemical mechanisms allowing this alga to tolerate such conditions are still largely unknown, as very few studies of intertidal seaweeds focus on *Pelvetia* [6,7]. These works and the studies of the mid and low intertidal fucoids (*Fucus vesiculosus*, *F. serratus*, *Ascophyllum nodosum*) suggest that fluctuations of physiological parameters and biochemical composition induced by the tidal cycle include changes in photosynthetic performance and pigment content [7,8,9], nutrient uptake [10], reactive oxygen species metabolism [7,11], osmolyte accumulation [1], and secondary metabolism [12]. Moreover, intertidal fucoids were shown to use an adaptive strategy similar to CAM-photosynthesis, storing CO_2_ in the form of organic acids during submersion and then consuming it in the Calvin cycle during the first hours of air exposure [13,14]. Phlorotannins, the dominating secondary metabolites of brown algae, also contribute to resilience of intertidal fucoids [15,16]. These phenolic compounds have considerable UV-protective and antioxidant capacities, and thus may provide chemical defense against high light and temperature stress during the low tide [12]. All these data imply that the physiological adjustment of *P. canaliculata* to the variation of environmental conditions imposed by the tidal cycle is accompanied with dramatic changes of cell metabolism, which remain obscure at the moment.

Here, we used a GC-MS-based metabolomics approach to explore the peculiarities of *Pelvetia* biochemistry in the context of the unique tolerance to recurring desiccation demonstrated by this species. This method allows simultaneous analysis of hundreds of low-molecular weight primary and secondary metabolites, which makes it an ideal method for comprehensive characterization of such potentially complicated processes. Metabolomic studies of brown macrophytes are still rare. The most detailed investigations were carried out on *Ectocarpus siliculosus* [17,18] and *F. vesiculosus* [19,20], whereas most other metabolite profiling studies focus on dominating metabolites and compounds having a potential applied relevance [21,22,23]. In our study, we discuss and relate the results of metabolite profiling of *Pelvetia* to data on relative water content, stress markers, titratable acidity, pigment, and phlorotannin content.

Therefore, the objective of the current study is to compare the biochemical composition of the thalli of *P. canaliculata* during the different phases of the tidal cycle. To our knowledge, this study is the first attempt to draft a detailed biochemical network underneath the complex physiological processes conferring the successful survival of this organism in the harsh conditions of the high intertidal zone of the polar seas.

## 2. Results

### 2.1. Water Content and Titratable Acidity in the Algal Thalli at Different Tidal Phases

Relative water content (RWC) in the thalli of *P. canaliculata* changed dramatically during the tidal cycle (Figure 1). When under water (HW and ET phases), the thalli had RWC of ~100%, but after the emersion, this parameter decreased by more than 3.5 times and then was stable during the whole period of air exposure (LW and RT phases). For the details of the local tidal regime and the sampling timepoints, please see Section 4.1 of Materials and Methods.

The changes in RWC of algal thalli were accompanied by the oscillations of titratable acidity (Figure 2). This parameter was maximal (~144 µmol H^+^ g^−1^ DW) during the underwater phases (HW and ET) and decreased significantly (*p* < 0.05) at the waterless phases (LW and RT).

### 2.2. Hydrogen Peroxide and Malondialdehyde Content in the Algal Thalli at Different Tidal Phases

Hydrogen peroxide accumulated in the cells of *P. canaliculata* to its maximum (0.19 µmol g^−1^ DW) soon after submersion (HW phase), then tended to decrease (though not significantly) till the ET phase and dropped abruptly after emersion of the thalli (LW phase) (Figure 3). This reduction in H_2_O_2_ content was accompanied by the increase in malondialdehyde (MDA) level, which was the highest (74 nmol g^−1^ DW) at the LW phase (Figure 4). In progression of the air exposure, there were no considerable changes in H_2_O_2_ and MDA content in *Pelvetia* cells (Figure 3 and Figure 4). After submersion, the MDA amount gradually decreased, reaching minimal values during the ET phase, when the alga spent ~4–4.5 h under water (Figure 4).

### 2.3. Pigment Content in the Algal Thalli at Different Tidal Phases

The most considerable changes in pigment content in the thalli of *P. canaliculata* during the tidal cycle related to the amounts of chlorophyll *a* and the product of its degradation, pheophytin *a* (Figure 5). Chlorophyll *a* content was maximal (0.8–0.9 mg g^−1^ DW) during the underwater phases (HW and ET) and dropped almost two-fold during the waterless phases (LW and RT). In contrast, the amount of pheophytin was the lowest (1.8% of summarized contents chlorophyll *a* and pheophytin *a*) at the end of the underwater phases (ET) and increased dramatically in progression of the air exposure.

Chlorophyll *c* contributed to 13% of the total chlorophyll amount during the HW phase and up to 23% during the LW phase. Generally, the tide-induced oscillations of the content of this additional chlorophyll were not as clear as those of chlorophyll *a*, though chlorophyll *c* also tended to accumulate in the submerged thalli, reaching its maximum at the ET phase (Figure 5). The total amount of carotenoids in the cells of *P. canaliculata* varied from 0.43 to 0.64 mg g^−1^ DW, gradually decreasing during the underwater phases (HW and ET) and increasing during the waterless phases (LW and RT) (Figure 5).

### 2.4. Phlorotannin Content in the Algal Thalli at Different Tidal Phases

Total phlorotannin content in the thalli of *P. canaliculata* at different tidal phases varied from ~4.5 to 7.4% DW (Figure 6). Notably, different tide-induced behavior was shown for the two subcellular pools of these phenolic metabolites: intracellular phlorotannins, located in physodes, and phlorotannins associated with the cell wall (CW). The amount of intracellular phlorotannins was highest in the air-exposed thalli (LW and RT phases) and sharply decreased after ~2–2.5 h of immersion (HW phase). The dynamics of CW-bound phlorotannin content was just the opposite: these molecules accumulated in the *Pelvetia* thalli under water (up to 15.6% of total phlorotannins) and had the lowest content at the LW phase, after 3.5–4 h of air exposure (Figure 6).

### 2.5. Metabolic Profiles in the Algal Thalli at Different Tidal Phases

GC-MS-based metabolite profiling of *P. canaliculata* revealed 132 compounds identified by retention indices and spectral similarity (excluding 110 unknowns), represented by 66 carbohydrates (sugars, polyols, and sugar acids), 17 amino acids including three non-proteinogenic ones (5-hydroxypipecolic acid, α-aminoadipic acid, ornithine), 11 organic acids, 9 fatty acids and their derivatives, 9 phenolic compounds (phloroglucinol and its dimers, homogentisic acid, salicylic acid etc.), and a miscellaneous group of metabolites including fucosterol, tocopherols, ascorbic acid, squalene, urea, etc. (Appendix A). The most abundant compounds in the metabolite profiles were mannitol, volemitol, citric acid, fucosterol, threonic acid, and several polyols and sugars, which could not be identified unambiguously.

The main changes of metabolite profiles of the *P. canaliculata* thalli during different tidal phases were revealed by a partial least squares discriminant analysis (PLSDA), where the first three components explain ~60% of the variance (Figure 7). The first component divides all the samples into two distinct groups, separating underwater phases (HW and ET) from waterless phases (LW and RT). Analysis of the component loadings showed that the metabolites contributing significantly to component 1 are mostly carbohydrates (fructose, myo-inositol, threonic acid), amino acids (threonine, glutamine, proline, etc.), phenolic compounds (phloroglucinol, 5-hydroxypipecolic, and homogentisic acids) and, to a lesser extent, organic acids (citric and isocitric acids) and components of lipid metabolism (cholesterol, glycerol-myristate). Component 2 allows separating HW and ET phases. The most significant constituents of this component are photosynthetic products, different sugars and polyols (glucose, sucrose, dulcitol), the amino acid phenylalanine, and several compounds related to lipid metabolism. The difference between the metabolite profiles of *Pelvetia* thalli during LW and RT phases was only revealed at the level of component 3 (Figure 7). Among the compounds showing high loadings for this component are salicylic acid, sugar acids (gluconic and galacturonic), several organic acids (malic, succinic), fatty acids, and squalene.

The specific ‘metabolite signatures’ of *Pelvetia* passing through different tidal phases are illustrated by a heatmap (Figure 8). During the HW phase, cells featured relatively low contents of carbohydrates, compounds related to lipid metabolism and phenolics, but accumulated free amino acids (except for phenylalanine) and several organic acids of TCA cycle (citric, isocitric, cis-aconitic, malic). Among phenolic compounds, the notable exception is homogentisic acid, which showed the highest relative content at this tidal phase. In progression of the underwater period (ET phase), the changes in the carbohydrate profiles occurred: several sugars and polyols were up-regulated (glucose, sucrose, polyol RI2858), whereas the content of the other compounds (inositols, dulcitol, threonic acid) decreased (Figure 8).

The emersion and transition to the air-exposed conditions lead to dramatic rearrangements of the *Pelvetia* metabolome. The most prominent features of metabolite profiles related to the LW phase are an accumulation of low molecular weight carbohydrates, phenolic compounds (including tocopherols), and dicarboxylic acids in *Pelvetia* cells, and a decrease in relative amounts of free amino acids and citric and isocitric acids (Figure 8, Figure 9 and Figure 10). In addition, homogentisic acid behavior is inconsistent with the other phenolic metabolites, demonstrating a significant (*p* < 0.05) decrease in relative content after the emersion of the algal thalli (Figure 8 and Figure 10).

During the RT phase, after 7–8 h of air exposure, fatty acids and squalene were considerably up-regulated together with several sugars (glucose, trehalose, sucrose). Meanwhile the contents of free amino acids and most organic acids (in particular, the tricarboxylic ones) kept decreasing, reaching the minimum values at this tidal phase (Figure 8 and Figure 9). While most of phenolic compounds maintained their relatively high level throughout both waterless tidal phases (LW and RT), the content of salicylic acid, after a sharp 10-fold increase during the first hours of air exposure, dropped down with progressing desiccation (Figure 8 and Figure 10). The only carbohydrate demonstrating a clear down-regulation at the RT phase was gluconic acid (Figure 8).

## 3. Discussion

In this study, we addressed, to our knowledge for the first time, the metabolic adjustment of the high intertidal brown alga *P. canaliculata* to the recurrent changes of environmental conditions during the tidal cycle. For all intertidal organisms derived from the sea, such as algae, the most crucial event is the emersion during the waterless period of the cycle [1]. For *Pelvetia*, these periods of desiccation last for at least 7–8 h, even during the spring tides and up to four days during the neap tides. From our data, we can see the dramatic changes in RWC in the *Pelvetia* cells (Figure 1). Notably, more than a three-fold decrease in RWC already occurs 3–3.5 h after emersion and then this parameter is stable for the next several hours in the air. The rehydration is also very fast, and, as known from the literature data, the complete recovery of key physiological processes took no more than 2 h, even after more prolonged and severe desiccation [24]. For most terrestrial plants, a decrease of RWC below 60% results in permanent wilting [25,26]; thus RWC dynamics in *Pelvetia* with regular drops to 28% (as in our study) or lower values (e.g., in the period of neap tides) resembles that of the poikilohydrous photosynthetic organisms, such as the resurrection plants or lichens [27,28]. Indeed, such extreme desiccation tolerance may arise from the mycophycobiosis state of *P. canaliculata* and ubiquitous presence of the endophyte fungus in its thallus [4]. This hypothesis is supported by another example of intimate association between a brown alga and a fungus: *Petroderma maculiforme* (Ishigeales) lichenized by the ascomycete *Wahlenbergiella tavaresiae* inhabits the high intertidal zone (the same biotope as *P. canaliculata*), whereas the free-living individuals of *P. maculiforme* typically grow in the low intertidal or even subtidal zone [29,30,31]. It is suggested that symbiosis with *W. tavaresiae* gave the alga higher resistance to desiccation and wave action and thus promoted the colonization of a new habitat [32].

The poikilohydrous nature of *P. canaliculata* has an important consequence from the perspective of the methodology used to study this organism. Considerable changes of thallus moisture make any measurements of its physiological or biochemical parameters calculated on the fresh weight (FW) basis inadequate, as it is frequently conducted for other algae or vascular plants. In our study, we calculated all the parameters per dry weight, while FW-based calculations were used in several earlier studies of intertidal seaweeds (e.g., [7,9]), which hampers data comparison.

The other specific feature of *Pelvetia*, which should be taken into account while discussing its biochemical composition, is the ubiquitous presence of the hyphae of the endophyte fungus in its thalli [4,5]. Though the total biomass of the fungus is several orders of magnitude lower than that of the alga, some especially abundant fungal metabolites may be detected by GC-MS analysis and contribute to the results. In this study, we did not find any metabolites in *Pelvetia* sample that were specific for fungi and not reported for algae.

Considering the repeating of waterless periods in the life of *Pelvetia* and its rapid full recovery after rehydration, we wondered whether air exposure is a true physiological stress for this organism, resulting in significant damage to cellular constituents. As we took the LW and RT samples when these tidal phases occurred in the daytime in warm and sunny weather, desiccation was accompanied by light and temperature impacts. Altogether these factors may lead to disbalance in reactive oxygen species (ROS) metabolism, oxidative stress, and loss of integrity of cell macromolecules and membranes. As hydrogen peroxide and MDA contents are widely accepted biochemical markers of oxidative stress and level of lipid peroxidation [33,34], we monitored these parameters in *Pelvetia* cells throughout the tidal cycle (Figure 3 and Figure 4). Both H_2_O_2_ and MDA content data measured in our study were within the range of the values reported for different macroalgae (including other populations of *P. canaliculata*) and aquatic vascular plants [7,35,36,37,38,39]. According to the literature data, in seaweeds, different abiotic stresses are accompanied by considerable changes in ROS production and MDA level. Thus, H_2_O_2_ production enhanced more than thrice in the thalli of *F. evanescens*, *F. distichus*, and *Scytosiphon lomentaria* after desiccation stress [9,40] and in *Ectocarpus siliculosus* after heavy metal stress [41]. MDA content was shown to increase by one and a half time in *F. vesiculosus* thalli exposed to enhanced temperature [39] and in *Macrocystis pyrifera* treated with high temperature and UV-light [42]. In the current study, there was a slight increase (by ~25%) of MDA level in *Pelvetia* cells after the emersion, but it did not grow further, either in progression of the waterless period, nor after rehydration (Figure 4), and thus can hardly be interpreted as a serious threat to cell membranes’ integrity. As for H_2_O_2_ content dynamics, surprisingly, *Pelvetia* cells accumulated more of this ROS when underwater, compared to the waterless period (Figure 3), and again the amplitude of these oscillations was not large enough to consider them as clearly stress-related. One of the main sources of H_2_O_2_ in the cells of plants and algae are photosynthetic processes [34], and rehydration of *Pelvetia* thalli is accompanied by a considerable increase in photosynthesis rate [6,24], which may account for the rise of H_2_O_2_ production in the submersed thalli.

The only clear evidence of damage to functional molecules in *Pelvetia* cells during the waterless tidal phases came from the analysis of the pigment content (Figure 5). The amount of chlorophyll *a* decreased almost two-fold after the emersion, and, moreover, the content of pheophytin increased dramatically to the end of the waterless period. Pheophytin is one of the first products of chlorophyll degradation, and its accumulation refers to either cell senescence or damage of the light-harvesting apparatus [43]. Relative pheophytin content was shown to increase in different algae and vascular plants subjected to osmotic, high light, and heavy metal stresses [44,45,46]. Notably, soon after immersion, pheophytin content in *Pelvetia* cells declines to a level not exceeding 5% of the sum of chlorophyll + pheophytin, thus showing the high sustainability of the pigment metabolism.

As a whole, analysis of the stress markers dynamics during the tidal cycle showed that 7–8 h long emersion even in combination with warm and sunny weather may be considered as a relatively mild stress, if any at all, for the White Sea populations of *P. canaliculata*. Apparently, the metabolic protection mechanisms in this alga (and, presumably, also in its fungus symbiont) are well tuned to preclude the dramatic changes of ROS level, lipid peroxidation intensity, or irreversible damage of photolabile molecules. Thus, it might be more correct to consider waterless periods of *Pelvetia* life not as a permanent stress, but as a specific metabolism state, characterized by appropriate biochemical rearrangements conferring the successful survival.

The results of PLSDA with evident separation of the HW+ET samples from the LW+RT samples at the level of the first component (Figure 7), clearly indicate that during the tidal cycle, *Pelvetia* passes through two distinctive metabolism states depending on the current environment: water or air. According to the literature data, very soon after the submersion and rehydration, *P. canaliculata* begins to actively photosynthesize [6,24], which corresponds well with our data showing the increase in the chlorophyll content between the RT and HW phases (Figure 5). As we cannot see a considerable accumulation of the early photosynthetic products, such as glucose, mannitol, and other low molecular weight carbohydrates in *Pelvetia* cells at the HW phase (Figure 8), we may suppose that these compounds are immediately consumed for the biosynthesis reactions. The spectrum of the synthesized products may include laminaran (specific brown algal storage polysaccharide) and the other structural and functional molecules, such as amino acids, proteins, and cell wall constituents. Relatively high amounts of free amino acids (glutamic acid, glutamine, aspartic acid, threonine, etc.) imply active efflux of ketoacids from the TCA cycle to form different amino acids and then proteins (Figure 8 and Figure 9). The most stable intermediates of TCA cycle (citric, isocitric, malic acids) are also present in relatively high concentrations indicating the active cell respiration. The drop of the intracellular phlorotannin content accompanied by increase in the CW-bound phenolics may be the result of the intensification of the thallus growth and CW formation at the underwater tidal phases. During biosynthesis of brown algal cell walls, intracellular phlorotannins are secreted into the forming apoplast where they cross-link with the alginate molecules, thus conferring the wall rigidification [38,47].

Apparently, at the end of the underwater period (ET phase), the photosynthetic machinery of *Pelvetia* gets optimized: the cells contain a minimal amount of pheophytin, the content of chlorophyll *c* increases, and contents of H_2_O_2_ and MDA tend to decrease (Figure 3, Figure 4 and Figure 5). Accumulation of glucose, together with the metabolically linked sugars (sucrose, mannose), may indicate the stabilization of the intracellular laminaran pool and slow its further replenishment (Figure 8). Interesting features of the metabolite profiles of the ET samples are a sharp increase in phenylalanine content and gradual accumulation of tocopherols (Figure 8 and Figure 10). These changes find their logical explanation in the subsequent events, connected with the emersion of *Pelvetia* thalli. Our data show that adaptation of *P. canaliculata* to air exposure includes extensive metabolic rearrangements (Figure 8, Figure 9 and Figure 10). Notably, one of the most affected compounds, transiently up-regulated (fold change > 10) at the beginning of the waterless period, was salicylic acid (Figure 8 and Figure 10). Salicylic acid is a biologically active compound of phenolic nature, known as a signal molecule, involved in the induction of diverse adaptive reactions to abiotic and biotic stresses in both vascular plants and algae [48,49,50]. Thus, in the red macrophyte *Neoporphyra haitanensis* expression level of the genes related to the salicylate biosynthesis and content of endogenous salicylic acid increased considerably after exposure to high temperature, high light, desiccation, and ultraviolet irradiation [49,51]. Moreover, pretreatment with salicylic acid was shown to improve thermotolerance of the brown and red seaweeds due to activation of the enzymes contributing to antioxidative defense and stabilizing the content of reactive oxygen species in cells [52,53]. Wang et al. [51] showed that in the cells of *N. haitanensis*, biosynthesis rate and content of salicylic acid increased sharply in an hour after beginning the exposure to high temperature, desiccation, and high light, but returned to the initial level after 6 h of treatment. This coincides with our data: in *P. canaliculate*, salicylic acid content grows dramatically in the beginning of the waterless period (LW phase), but then drops down in progression of the air exposure (RT phase) (Figure 8 and Figure 10). Thus, we may suggest, that the spike of this phytohormone is a signal activating protective mechanisms conferring *Pelvetia* resilience during the waterless part of its life. To our knowledge, in our study, the endogenous content of salicylic acid in *P. canaliculata* was analyzed for the first time. The values measured in the underwater samples (12–16 nmol g^−1^ DW) are relatively high and comparable to the highest values reported for the other macroalgae, including the intertidal ones [54,55]. It is possible that the enhanced level of salicylate biosynthesis is one of the biochemical adaptations of *Pelvetia* to the high intertidal habitat. Salicylic acid can be synthesized via two independent pathways, operated by isochorismate synthase or phenylalanine ammonia lyase (PAL) as the key enzymes. Both these routes are supposed to occur in the vascular plants and algae, though their relative contributions to the level of this hormone in the cells vary considerably in different species [51,56,57]. The key intermediate of PAL-dependent biosynthesis of salicylic acid is phenylalanine, and in *P. canaliculata*, accumulation of this amino acid precedes the spike of salicylate during the tidal cycle (Figure 8 and Figure 10). Thus, we suggest that in *Pelvetia* salicylic acid is mainly formed via the PAL-dependent pathway.

Besides the salicylate precursor, during the underwater period, *Pelvetia* cells also accumulated the metabolically linked precursors of tocopherols, such as tyrosine and homogentisic acid (Figure 10). A considerable down-regulation of these metabolites at the LW phase coincided with the increase in tocopherols content. Due to their potent antioxidant capacities, tocopherols may contribute to the prevention of the oxidative stress in the air-exposed *Pelvetia* thalli. The other phenolic compounds, both low molecular weight (phloroglucinol, ditertbutylphenol, etc.) and polymeric (intracellular phlorotannins), also accumulated in *Pelvetia* cells during the waterless tidal phases (Figure 6 and Figure 8). These metabolites are known for their UV-protective and radical-scavenging activities [16,58]. The increase in amounts of tocopherols and phenolic compounds was earlier shown in *F. vesiculosus* thalli exposed to enhanced temperature [59]. Notably, only the intracellular fraction of phlorotannins, but not the CW-bound one, increased in the air-exposed *Pelvetia* cells (Figure 6). This corresponds well to the results of our previous studies, where we showed that these two subcellular pools of phlorotannins differ in their molecular composition, and that free radical scavenging activity of the CW-bound phlorotannins of *P. canaliculata* is substantially lower compared to that of the intracellular phlorotannins [60,61]. Generally, among studied fucoid algae, *P. canaliculata* is distinguished by its unusually complex phlorotannin profile, which was suggested to reflect both adaptation of this species to high intertidal conditions and interaction with the endophyte fungus [60,62].

The adaptive responses of plants and algae to desiccation typically include the up-regulation of osmolytes, such as soluble sugars and polyols. In *Pelvetia*, we could also see the accumulation of diverse low-molecular weight carbohydrates (~2–9-fold for a broad range of identified and unknown sugars and polyols) during the waterless tidal phases. Interestingly, one of the most affected carbohydrate derivatives was gluconic acid, showing a spike at the LW phase (Figure 8). The biological function of this metabolite in plants and algae is still very poorly studied, though there are several reports of its up-regulation in response to different stresses as well as to exogenous salicylic acid treatment in vascular plants [63,64,65,66]. We suggest that the spike of gluconic acid in *Pelvetia* cells may indicate a transient switch of glucose catabolism from glycolysis to the oxidative pentose phosphate pathway (PPP), which was shown to activate under oxidative stress conditions and to contribute to stabilization of the cell redox status [67,68]. Gluconic acid can be produced due to glucose oxidation by NADP^+^ glucose dehydrogenase in the gluconate shunt, an alternative route for the entrance of glucose into the PPP, which was shown to function in plants and algae [69]. According to our data, increase in gluconate content coincided with the transient drop of glucose level (Figure 8).

The biochemical rearrangements in the emersed *Pelvetia* thalli included decrease in free amino acid content and gradual accumulation of lipid exchange metabolites. Such events may be explained by the changes in the TCA cycle configuration, the more that relative amounts of the organic acids also changed dramatically after emersion (Figure 8). It is known that intermediates of the TCA cycle such as citric, α-ketoglutaric, and oxaloacetic acids can act as “switch valves” for the efflux of metabolites from the cycle for various biosynthetic processes [70]. Apparently, in the submerged thalli, there was no considerable efflux of citric acid from the cycle; meanwhile, α-ketoglutarate and oxaloacetate were largely consumed for amino acid biosynthesis. On the contrary, after the emersion, citric acid might start to leave the cycle providing material for the synthesis of phlorotannins, squalene, and fatty acids, resulting in a two-fold decrease in total citrate content (Figure 9). Both fatty acids and squalene are membrane constituents, so they may be used to maintain cell membranes’ integrity during desiccation of the thalli. The spikes of succinate and fumarate at the LW phase may be the result of limited flow of tricarboxylic acids and general inhibition of cell respiration occurring during the waterless period [24,71]. One more organic acid found in *Pelvetia* extracts (citramalate) co-behaved with succinate and fumarate (Figure 8). This metabolite, previously only described in microorganisms, was recently found in several plants, brown algae, and filamentous fungi [72,73,74]. In apples, citramalate is supposed to be synthesized from pyruvate and acetyl-CoA by citramalate synthase and to contribute to isoleucine and 2-methylbutanoate and propanoate ester biosynthesis [75]. The biochemical role of this compound in brown algae is still to be investigated. As citramalate may be a fungal metabolite, in *Pelvetia* it may also be produced by the other member of mycophycobiosis, *S. ascophylli*.

Citric acid is one of the predominant metabolites of brown algae with cellular concentrations several orders of magnitude higher than the other organic acids (Appendix A; [18,20]). Apparently, this compound, standing on the crossroads of several metabolic pathways, is largely involved in the metabolic adjustment of *Pelvetia* to extreme conditions. One of its putative functions may be the carbon storage and recycling in the processes similar to CAM-photosynthesis. According to the literature data, another intertidal alga, *F. vesiculosus*, can photosynthesize for some time even in CO_2_-free air, which suggests the presence of an organic carbon store in the cells [13,14]. Moreover, *Fucus* and *Pelvetia* not only maintain photosynthesis during the first hours of emersion, but even demonstrate a transient increase in CO_2_ fixation rate, despite the lower inorganic carbon concentration in the air compared to seawater [24,71,76]. As measurements of tissue titratable acidity during the periods differing in inorganic carbon and water availability (day vs. night, for terrestrial CAM plants) were shown to be a highly sensitive method for detecting even low-level CAM activity [77], we carried out such analysis and revealed considerable changes of H^+^ content in emersed and submersed *Pelvetia* thalli (Figure 2). The amplitude of these oscillations is comparable to the values reported for some terrestrial CAM plants (e.g., *Pilea peperomioides*, *Werauhia sanguinolenta*) [77], and citric acid is the only metabolite present in *Pelvetia* cells in the amount high enough to account for such acidity changes. Stoichiometrically, each mol of citrate accumulated in algal cells during the underwater period should be balanced by 3 mol H^+^ during titration, and indeed the ratio of the amplitudes of citrate and H^+^ oscillations, measured in our study, is very close to 1:3 (exactly, 1:2.97) (Figure 1 and Figure 9). Though the principal organic acid involved in typical CAM-photosynthesis is malate, considerable diurnal oscillations of citrate level were shown for many terrestrial CAM plants (reviewed in: [78]). E.g., during the salinity-induced C_3_–CAM shift in *Mesembryanthemum crystallinum*, a nocturnal accumulation of citrate preceded that of malate, and the CAM-dependent increase in citric acid content was ~100 times higher than that of malic acid [79]. Currently, the biochemical and ecophysiological significance of these CAM-associated citrate oscillations have been studied very little. Lüttge [78] suggested that using citric acid for carbon storage in plant cells may be reasonable and even favorable if it is synthesized from acetyl-CoA obtained not from glycolysis (as it would mean the loss of CO_2_ due to pyruvate decarboxylation), but from other sources, such as β-oxidation of fatty acids. Such a process implies oscillations of fatty acid levels in counter-phase to those of citrate, and this is exactly what we found in the tide-dependent dynamics of *Pelvetia* metabolite profiles (Figure 8 and Figure 9). The “open”, non-cyclic, configuration of TCA cycle may provide export of citrate to the cytosol and/or vacuole, and its decarboxylation (via isocitrate) may occur then due to activity of cytosolic NADP-isocitrate dehydrogenase [79]. Thus, we suggest that besides providing organic carbon for biosynthetic reactions, citric acid may also serve as a CO_2_ store and, thus, contribute to maintaining the photosynthetic activity in the emersed *Pelvetia* thalli. To our knowledge, our study provided the first data implying that citrate-based CAM-like processes may occur in brown algae. This issue definitely warrants further investigation.

## 4. Materials and Methods

### 4.1. Algal Material Collection

Thalli of *Pelvetia canaliculata* (L.) Dcne and Thur. were collected at the rocky shores of Srednii island in the Keret Archipelago (Kandalaksha Bay, White Sea) in July–August 2021–2022. This location features semi-diurnal tides with typical ranges of 1.0 m at neap tides and 1.8 m at spring tides. The material collection was carried out at four timepoints corresponding to different tidal phases (Table 1). To minimize the effects of varying environmental conditions, the sampling was made only in the daytime, in quiet and sunny weather and the air temperature was in the range of 20–25 °C. Most samples were taken during the spring tides. As at the neap tidal cycles *Pelvetia* cannot be regularly submerged, the thalli were not collected during this period.

Algae collected during high water or ebb tide were delivered to the laboratory in seawater. Algae collected during low water or rising tide were transported in dry condition. All fixations and analyses were started no later than 0.5 h after the sampling.

### 4.2. Fresh Weight, Dry Weight, and Relative Water Content (RWC) Determination

RWC was calculated according to [80]. For the determination of fresh weight, 12–20 samples, taken from different individual thalli during each tidal phase, were blotted with filter paper (for HW and ET samples) and weighed. For determination of dry weight, the samples were weighed again after drying at 70 °C to the constant weight. For determination of the saturated weight (SW), *Pelvetia* thalli were kept in water for 3 days and then blotted with filter paper and weighed. After that, RWC was calculated as $\frac{\left(FW-DW\right)\times 100\%}{SW-DW}$ [80].

Because of high variation of moisture between the samples taken at different tidal phases, all measured biochemical parameters were calculated on the DW basis.

### 4.3. Hydrogen Peroxide Analysis

Measurement of H_2_O_2_ content in thalli of *P. canaliculata* was carried out using the ferrous–xylenol orange assay (FOX) [81,82] with modifications. From 50 to 150 mg FW of algal material was homogenized in cold 0.2 M perchloric acid and the extracts were centrifuged (10,000× *g*, 5 min, 4 °C). The supernatants were neutralized with KOH and then treated with ascorbate oxidase (0.25 mL of sample, 0.75 mL of 0.1 M potassium phosphate buffer pH 5.6, 1 U of ascorbate oxidase, Sigma-Aldrich, Taufkirchen, Germany, A0157) for 10 min. Then each extract was divided into two aliquots, one of which was treated with catalase (Sigma-Aldrich, Taufkirchen, Germany, C9322). Equal volumes of FOX reagent (0.2 mM xylenol orange, 200 mM sorbitol, 50 mM H_2_SO_4_, 0.5 mM (NH_4_)_2_SO_3_, and 0.5 mM FeSO_4_) was added to both solutions. After 30 min, the extinction of the solutions was measured at 560 nm using a SPEKOL 1300 spectrophotometer (Analytik Jena AG, Jena, Germany). H_2_O_2_ concentration in the reaction mixture was calculated as the difference in absorbance between catalase-free and catalase-treated aliquots according to the calibration curve.

### 4.4. Malondialdehyde Analysis

Malondialdehyde (MDA) content was determined based on the protocol of Velikova et al. [83]. Samples of algal material (40 mg FW) were homogenized in cold 5% trichloroacetic acid (TCA), and the extracts were centrifuged (10,000× *g*, 5 min, 4 °C). The supernatants were transferred into 5 mL polypropylene tubes and supplemented with a 3-fold volume of 0.5% thiobarbituric acid (TBA, Sigma-Aldrich, Taufkirchen, Germany, T5500) in 20% TCA. The reaction mixture was incubated at 95 °C for 30 min and then cooled on ice. The extinction of the solutions was measured at 532 and 600 nm using a SPEKOL 1300 spectrophotometer (Analytik Jena AG, Jena, Germany). The content of TBA-reactive substances was calculated according to the published equations and expressed as MDA equivalent [83].

### 4.5. Pigment Analysis

Content of photosynthetic pigments in the thalli of *P. canaliculata* was measured spectrophotometrically. Fragments of algal thalli (5–10 mg FW) were ground using a mortar and pestle in aqueous acetone (90% for chlorophylls *a* and *c* and pheophytin *a* or 80% for total carotenoids) with a small amount of MgCO_3_. Several rounds of extraction were completed with additional acetone until the extracts were colorless. The pigment content was calculated according to the published equations [84,85,86] from data obtained with a SPEKOL 1300 spectrophotometer (Analytik Jena, Jena, Germany).

### 4.6. Determination of Total Phlorotannin Content

Extraction of intracellular and CW-bound phlorotannins was carried out based on the standard protocol of Koivikko et al. [87,88] with modifications described in [60]. Briefly, 20 mg of fresh algal material was poured with 70% aqueous acetone, homogenized, and left soaking in 1 mL aqueous acetone for one hour to extract intracellular phenolics. Then, the extract was centrifuged (5000× *g*, 10 min), the supernatant was transferred to another tube, and the pellet was re-extracted with another portion of aqueous acetone. The supernatants of four extraction rounds were combined.

The CW-bound phlorotannin fraction was extracted from the precipitate of the remaining algal material after the extraction of intracellular phlorotannins. The precipitate was resuspended in 0.5 mL of 1 M aqueous NaOH solution (80 °C) and then incubated for 2.5 h at room temperature with continuous shaking (750 rpm). After centrifugation (5000× *g*, 10 min,), the supernatant was transferred to another tube. The alkaline extraction was repeated three times. The combined supernatants were neutralized with concentrated HCl to pH 6.8–7.0.

A modification of the Folin–Ciocalteu micro-method was used to measure the total phenolic content in the extracts [89]. The reaction mixture containing 0.3 mL of sample (diluted as needed), 0.3 mL of Folin reagent and 2.4 mL of 5% (*w*/*v*) Na_2_CO_3_, was incubated for 20 min at 45 °C, and then the absorbance was measured at 750 nm using a SPEKOL 1300 spectrophotometer. Phloroglucinol (Sigma-Aldrich, Taufkirchen, Germany, 79330) was used as the standard.

### 4.7. Determination of Titratable Acidity

Samples of *Pelvetia* thalli (1–1.5 g FW) were fixed with boiling distilled water and homogenized. The homogenates were centrifuged (5000× *g*, 10 min), the supernatants were collected, and the pellets were re-extracted with another portion of boiling water. The supernatants of three extraction rounds were combined and cooled to 20 °C. Content of H^+^ in the extracts was determined by titration with 5 mM NaOH to pH 7.0.

### 4.8. Metabolite Profiling

Before starting the sample preparation, feasibility and appropriate sample amount were assessed in pilot experiments with ascending amounts of algal material. Fragments of algal thalli (10–20 mg FW, depending on the tidal phase) were poured with cold methanol (−25 °C), quickly ground in a pre-cooled mortar and left soaking in 1 mL of cold methanol for extraction. Aliquots of 500 μL of methanol extracts were transferred to clean 1.5 mL polypropylene Eppendorf tubes (VWR, Dresden, Germany) and vacuum-dried in the CentriVap vacuum concentrator system (Labconco, Kansas City, MO, USA) for subsequent analysis.

Metabolite profiling analyses were carried out according to [90]. Briefly, vacuum-dried extracts were incubated by shaking in methoxyamine hydrochloride (Alfa Aesar by Thermo Fisher Scientific, Kandel, Germany) solution in pyridine (Sigma-Aldrich, Taufkirchen, Germany) and *N,O*-bis(trimethylsilyl)-trifluoroacetamide (Macherey-Nagel GmbH and Co KG, Düren, Germany). After derivatization, samples were transferred to glass vial micro-inserts and subjected to GC-MS analysis on an Agilent 6890 gas chromatograph coupled to an Agilent 5973N quadrupole mass selective detector (Agilent Technologies, Böblingen, Germany) with standard electron impact ionization (70 eV). Separation was accomplished on a DB-5MS Ultra Inert column (Agilent, Waldbronn, Germany; 30 m × 0.25 mm ID and 0.25 µm film) at 0.9 mL/min carrier gas flow (He 5.0 Alphagaz, Air Liquide, Germany) after splitless injection at 250 °C. Within each sequence, a mixture of alkanes (C10–C32) in hexane was measured for the calculation of Kovats retention indices (RI). A mix of authentic reference standards containing 21 amino acids, 23 sugars and polyols, 19 organic acids, phloroglucinol, etc., was co-spiked to confirm the identity of expected compounds.

Peak deconvolution was accomplished using AMDIS 2.66. The retention indices were automatically calculated using an AMDIS calibration file containing the batch retention times of each alkane. GMD (Golm metabolome database, GMD_20100614_VAR5_ALK, 24 September 2010, [91]) and NIST14 (National Institute of Standards and Technology, Gaithersburg, MD, USA) were used for identification of the peaks based on spectra comparison. Where applicable, absolute quantitation was performed by standard addition using five calibration levels.

### 4.9. Data Analysis

Experiments were carried out with 5 to 20 biological replicates (taken from different individuals). Quantitation of metabolites in GC-MS analysis was performed by peak integration of the corresponding extracted ion chromatograms (*m*/*z* ± 0.5) for representative intense signals at specific retention times (RT) using Xcalibur 3.0. Excel 2013 (Microsoft, Redmond, WA, USA) and MetaboAnalyst 5.0 Web application (http://www.metaboanalyst.ca, accessed on 23 May 2023) were used for data processing and normalization procedures, creation of figures, and heatmap construction [92]. The metabolomic data processing included peak area normalization to the median of all areas within the corresponding chromatogram, generalized logarithm transformation, and range data scaling (mean-centered and divided by the range of each variable). The normalized metabolomics data were analyzed by a partial least square discriminant analysis (PLSDA). Where appropriate, data were confirmed for normality using the Kolmogorov–Smirnov test and for homogeneity of variance using the Levene test. Student’s *t*-test was used to confirm the significant differences between the means. All values are expressed as means and standard deviations.

## 5. Conclusions

During the underwater period, various growth-maintaining biosynthetic processes activate in the cells of *P. canaliculata*, which manifest as accumulation of chlorophyll, amino acids, and CW-bound phlorotannins. Emersion resulted in a dramatic decrease in RWC in *Pelvetia* thalli, though such severe desiccation did not lead to considerable increase in hydrogen peroxide and malondialdehyde content in the cells. Metabolic adjustment of *P. canaliculata* to emersion included accumulation of soluble carbohydrates, various phenolic compounds, including intracellular phlorotannins, and fatty acids. Apparently, the dominating metabolite of *Pelvetia*, citric acid, has a key role in switching the algal metabolism between the underwater and waterless conditions. Changes in titratable acidity underlain by the oscillations of citric acid content imply that some specific CAM-like processes may be involved in *Pelvetia* adaptation to the tidal cycle.

## Figures and Tables

**Figure 1 ijms-24-10626-f001:**
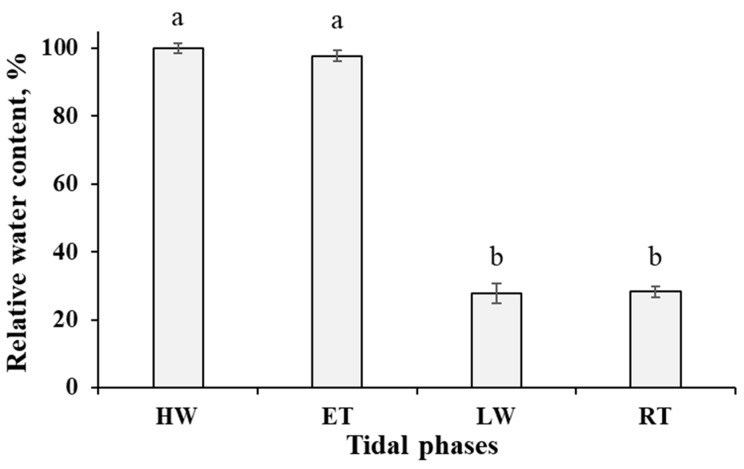
Dynamics of relative water content in the thalli of *Pelvetia canaliculata* during the tidal cycle. HW—high water; ET—ebb tide; LW—low water; RT—rising tide. Means are given with ±SD (*n* = 12–20); different letters indicate significant differences (*p* < 0.05, Student’s *t*-test).

**Figure 2 ijms-24-10626-f002:**
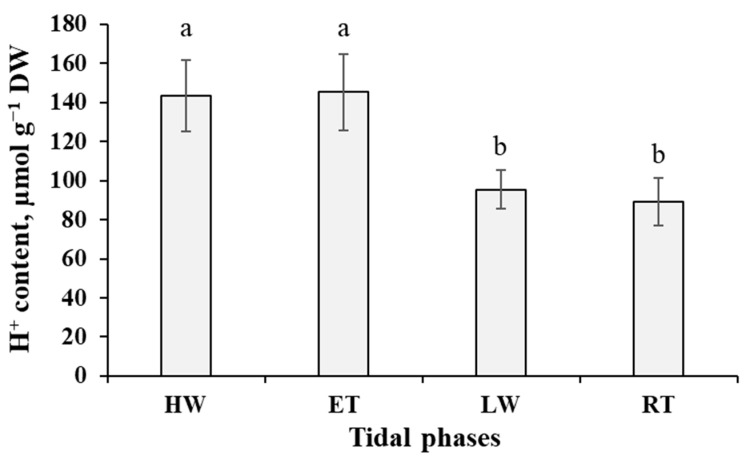
Dynamics of titratable acidity of the thalli of *Pelvetia canaliculata* during the tidal cycle. HW—high water; ET—ebb tide; LW—low water; RT—rising tide. Means are given with ±SD (*n* = 10–12); different letters indicate significant differences (*p* < 0.05, Student’s *t*-test).

**Figure 3 ijms-24-10626-f003:**
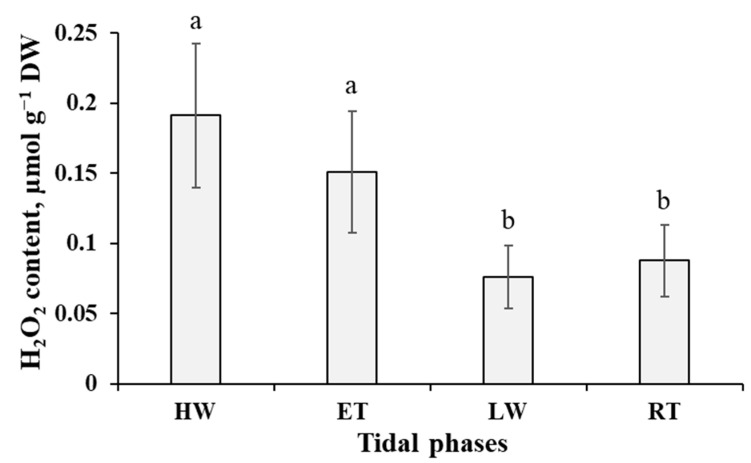
Dynamics of hydrogen peroxide content in the thalli of *Pelvetia canaliculata* during the tidal cycle. HW—high water; ET—ebb tide; LW—low water; RT—rising tide. Means are given with ±SD (*n* = 8–10); different letters indicate significant differences (*p* < 0.05, Student’s *t*-test).

**Figure 4 ijms-24-10626-f004:**
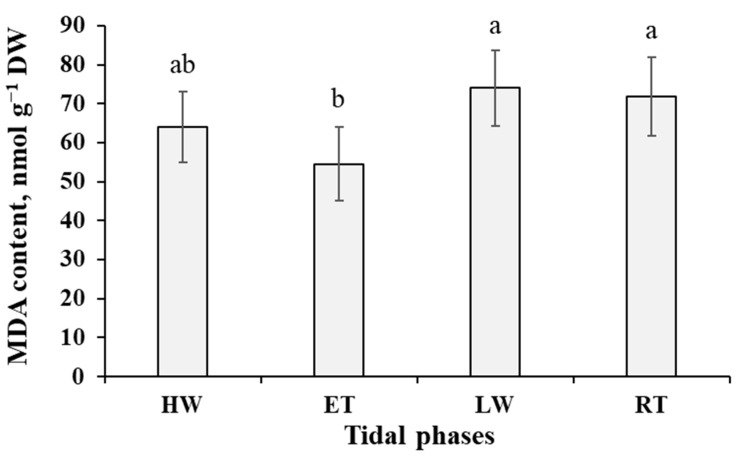
Dynamics of malondialdehyde (MDA) content in the thalli of *Pelvetia canaliculata* during the tidal cycle. HW—high water; ET—ebb tide; LW—low water; RT—rising tide. Means are given with ±SD (*n* = 6); different letters indicate significant differences (*p* < 0.05, Student’s *t*-test).

**Figure 5 ijms-24-10626-f005:**
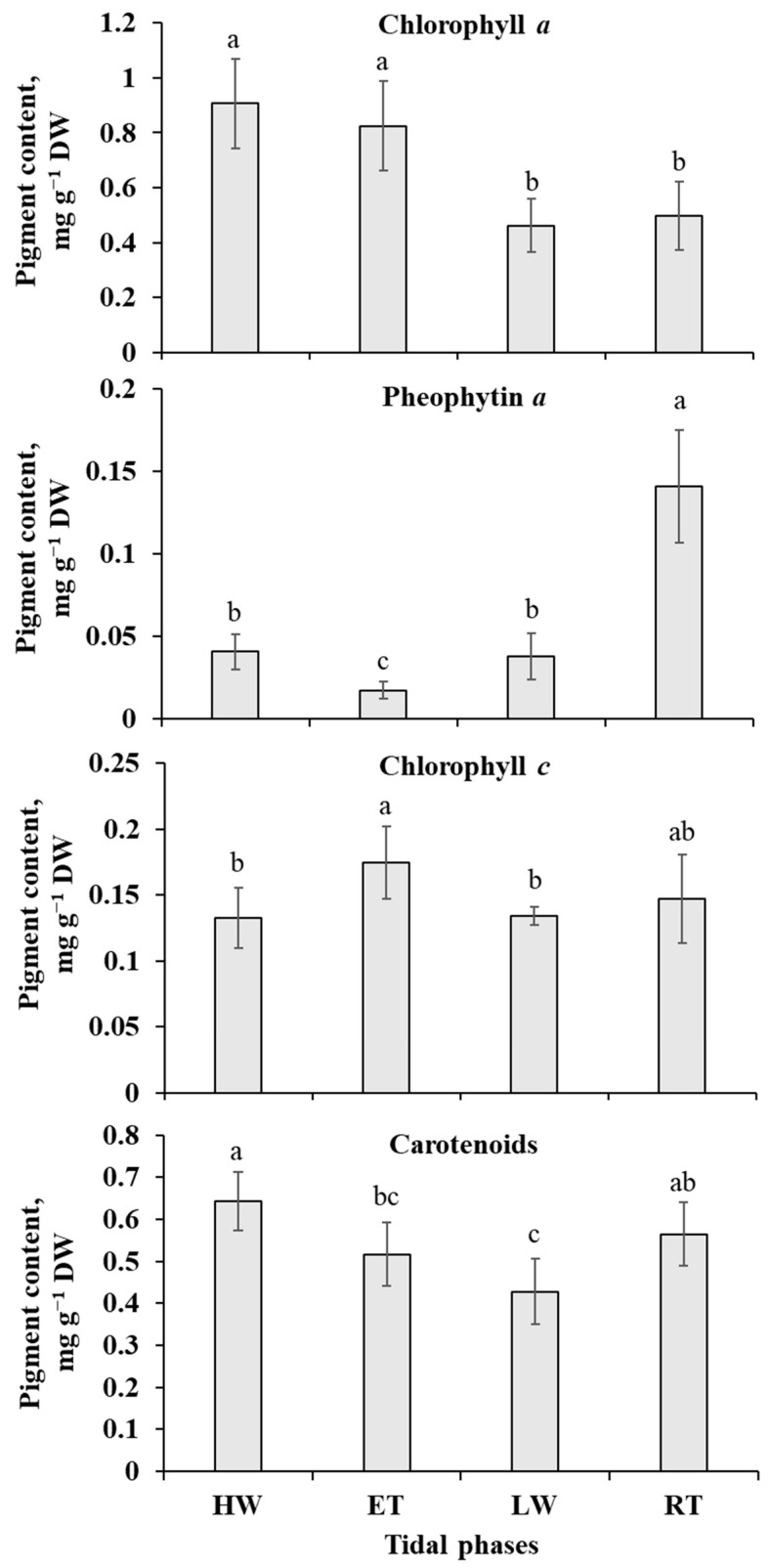
Dynamics of photosynthetic pigments content in the thalli of *Pelvetia canaliculata* during the tidal cycle. HW—high water; ET—ebb tide; LW—low water; RT—rising tide. Means are given with ±SD (*n* = 6); different letters indicate significant differences (*p* < 0.05, Student’s *t*-test).

**Figure 6 ijms-24-10626-f006:**
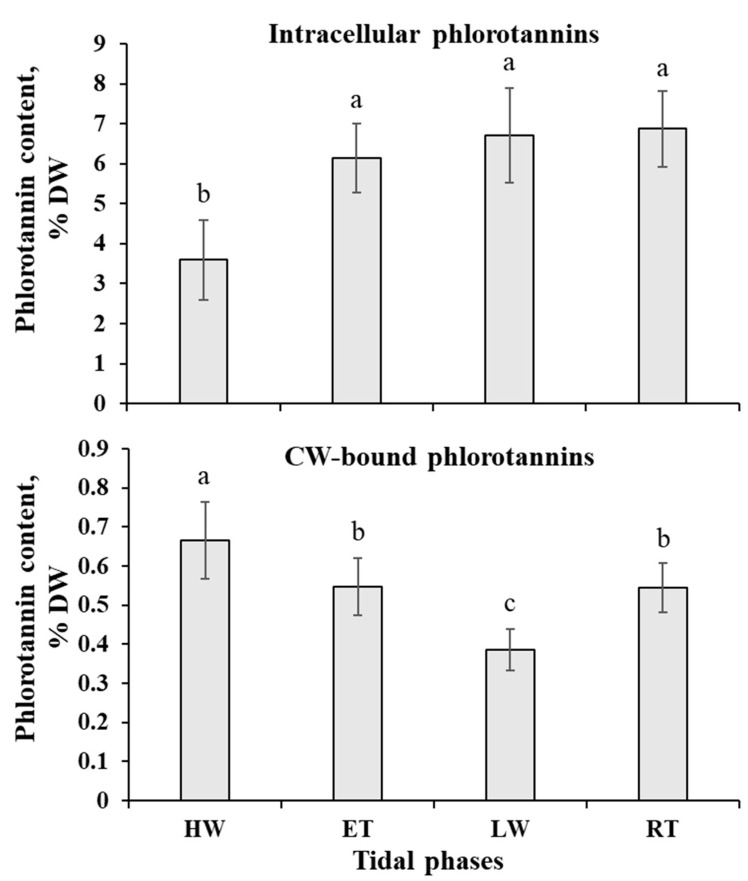
Dynamics of intracellular and cell wall (CW)-bound phlorotannin content in the thalli of *Pelvetia canaliculata* during the tidal cycle. HW—high water; ET—ebb tide; LW—low water; RT—rising tide. Means are given with ±SD (*n* = 8–12); different letters indicate significant differences (*p* < 0.05, Student’s *t*-test).

**Figure 7 ijms-24-10626-f007:**
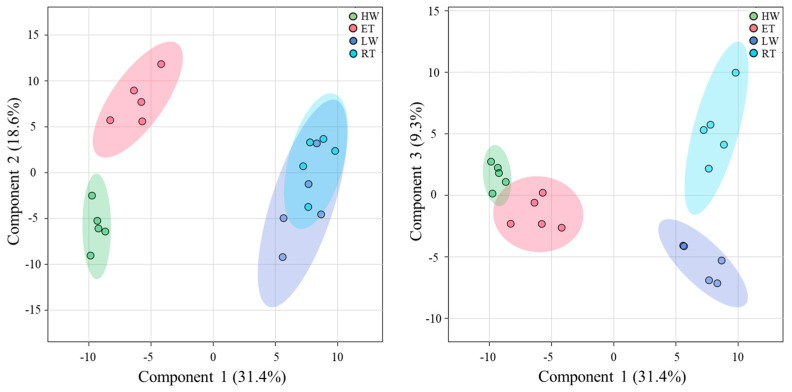
Sample scores for the first three components derived from PLSDA of the relative metabolite concentrations in the thalli of *Pelvetia canaliculata* during the tidal cycle. HW—high water; ET—ebb tide; LW—low water; RT—rising tide.

**Figure 8 ijms-24-10626-f008:**
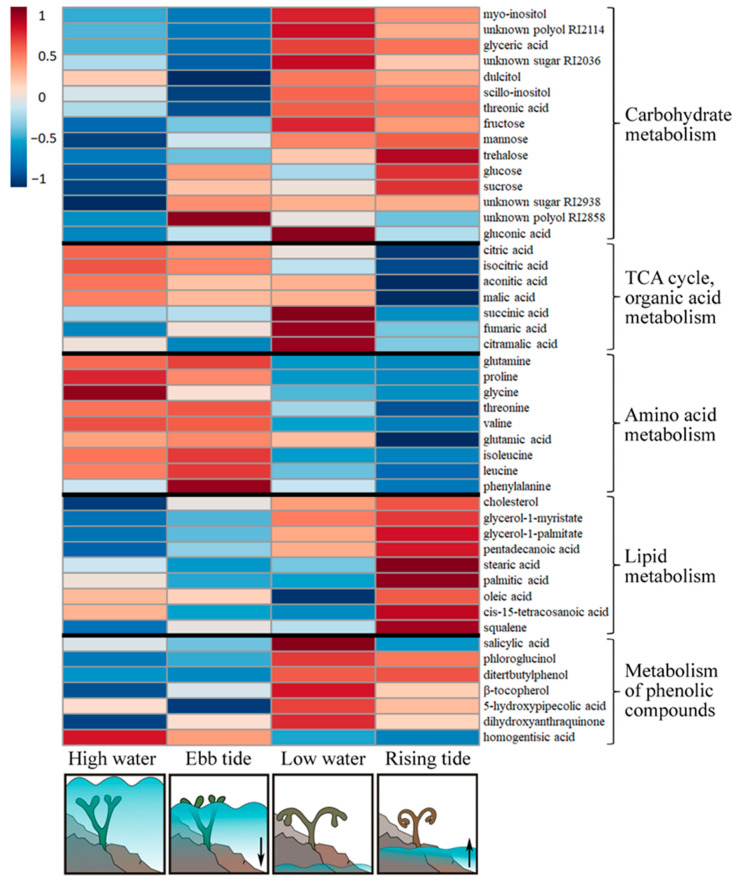
A heatmap of significantly (*p* < 0.05) changing key metabolites detected in the thalli of *Pelvetia canaliculata* during the tidal cycle. Mean area values of five samples are presented on a log_10_ scale. Arrows mean direction of the tidal current.

**Figure 9 ijms-24-10626-f009:**
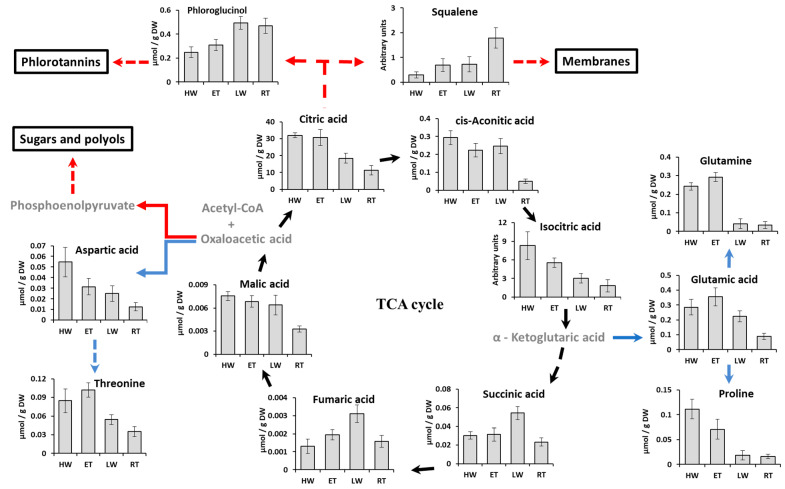
Presumable consumption pathways of TCA cycle metabolites in the cells of *Pelvetia canaliculata* during different tidal phases. Blue lines indicate the pathways activated in the water (high water, HW and ebb tide, ET), red lines indicate the pathways activated in the air (low water, LW and rising tide, RT). Direct reactions are presented as straight lines, and reactions involving several steps are presented as dashed lines. Metabolites which were not determined are labeled in grey. Bars represent the means ± SD. Arbitrary units are normalized peak areas of extracted ion chromatograms.

**Figure 10 ijms-24-10626-f010:**
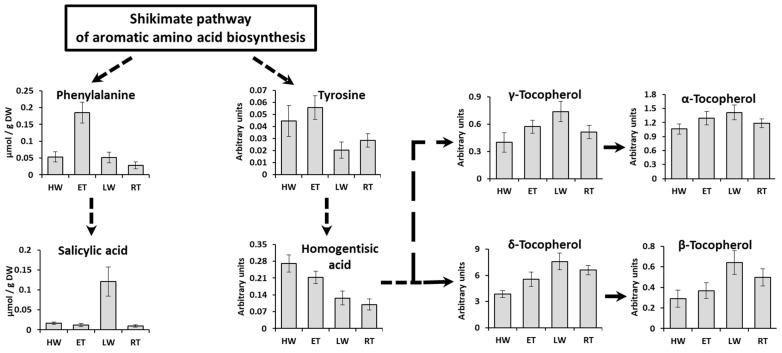
Scheme of the salicylic acid and tocopherols biosynthesis. Direct reactions are presented as straight lines and reactions involving several steps are presented as dashed lines. HW—high water, ET—ebb tide, LW—low water, RT—rising tide. Bars represent the means ± SD. Arbitrary units are normalized peak areas of extracted ion chromatograms.

**Table 1 ijms-24-10626-t001:** Sampling timepoints and condition of the thalli of *Pelvetia canaliculata* during the typical tidal cycle in the period of spring tides at the rocky shores of Srednii island.

Timepoint Label	Timepoint Description	Condition of *Pelvetia* thalli in the Moment of Sampling
High water (HW)	The maximum water level, no considerable tidal current	Totally submersed
Ebb tide (ET)	2–2.5 h after the water level started to decrease, tidal current is flowing seaward	Still under water, ready to emerge
Low water (LW)	The minimum water level, no considerable tidal current	In the air (exposure time: 3.5–4 h)
Rising tide (RT)	3.5–4 h after the water level started to increase, tidal current is flowing inland	Still in the air (exposure time: 7–8 h), ready to be flooded

## Data Availability

The datasets generated and/or analyzed in this study are available from the corresponding author upon reasonable request.

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
