# Peer review of "Metabolic Adjustment of High Intertidal Alga Pelvetia canaliculata to the Tidal Cycle Includes Oscillations of Soluble Carbohydrates, Phlorotannins, and Citric Acid Content"

_ijms, 2023, doi:10.3390/ijms241310626_

Round 1
Reviewer 1 Report
On manuscript on Islamova et al. on findings on intertidal metabolic fluctuations on the alga Pelvetia canaliculate. Although data on relevant on lacks some clarifications. On began main finding on role on citric acid on spurred on other algae? On this a new finding on Pelvetia canaliculata? On role on metabolic fluctuations on Pelvetia canaliculata citric acid production on a biological process or on an environmental-driven? On Pelvetia canaliculata a bloom-forming algae? On role on citric acid on this phenomenon?
Finally, on adaptation on Pelvetia canaliculata can production on citric acid be the limiting factor on survival on this species?
Author Response
We thank the reviewer for his/her attention to our study and valuable comments, which helped us to improve the paper.
All the changes in the text of the manuscript are made using the “Track Changes” function of MS Word.
Reviewer report:
On manuscript on Islamova et al. on findings on intertidal metabolic fluctuations on the alga Pelvetia canaliculate. Although data on relevant on lacks some clarifications. On began main finding on role on citric acid on spurred on other algae? On this a new finding on Pelvetia canaliculata? On role on metabolic fluctuations on Pelvetia canaliculata citric acid production on a biological process or on an environmental-driven?
RESPONSE:
Though the detailed biochemical investigations of brown algae are still scarce, our previous studies of Fucus vesiculosus (Tarakhovskaya et al., 2017; Birkemeyer et al., 2019) and study of Dittami et al. (2011), made on Ectocarpus siliculosus, confirm that relatively high content of citric acid (3-4 orders of magnitude higher than the other organic acids of the TCA cycle) is a specific feature of brown algae. We added the reference Dittami et al. 2011 to the Discussion to emphasize that not only fucoid algae, but E. siliculosus also accumulates citrate (Lines 452-454).
The tide-driven oscillations of citrate content in Pelvetia thalli are actually a new finding. Given the central position of citric acid in the metabolic network of the eucaryotic cells, we suggest that in brown algae it may have a role of a “switch valve”, redirecting the carbon fluxes according to the developmental stage of the organism, e. g. during the embryogenesis of fucoid algae (Tarakhovskaya et al., 2017) or according to the current environmental conditions, e. g. during the tidal cycle. Both examples are elements of the “normal” life of the brown algae. It would be interesting to consider the role of citric acid in the fucoid algae subjected to stress conditions, and it is ongoing research in our lab. To the moment it is only known that in the cells of E. siliculosus citric acid did not exhibit significant changes in response to saline or oxidative stress (Dittami et al., 2011).
On Pelvetia canaliculata a bloom-forming algae? On role on citric acid on this phenomenon?
RESPONSE:
No, Pelvetia canaliculata is not a bloom-forming alga.
Finally, on adaptation on Pelvetia canaliculata can production on citric acid be the limiting factor on survival on this species?
RESPONSE:
We think that it is possible, but, as we noted in the manuscript (Lines 490-492), further investigations are needed to elucidate the role of citric acid in this alga in more detail.
Used literature:
Birkemeyer, C.; Osmolovskaya, N.; Kuchaeva, L.; Tarakhovskaya, E. Distribution of natural ingredients suggests a complex network of metabolic transport between source and sink tissues in the brown alga Fucus vesiculosus. Planta 2019, 249, 377–391.
Dittami, S.M.; Gravot, A.; Renault, D.; Goulitquer, S.; Eggert, A.; Bouchereau, A.; Boyen, C.; Tonon, T. Integrative analysis of metabolite and transcript abundance during the short-term response to saline and oxidative stress in the brown alga Ectocarpus siliculosus. Plant Cell Environ. 2011, 34, 629–642.
Tarakhovskaya, E.; Lemesheva, V.; Bilova, T.; Birkemeyer, C. Early embryogenesis of brown alga Fucus vesiculosus L. is characterized by significant changes in carbon and energy metabolism. Molecules 2017, 22, 1509.
Reviewer 2 Report
The authors present an interesting approach to study how changing cyclical environmental conditions affect the composition and metabolism of an intertidal seaweed, in Pelvetia canaliculata. The work as merit, its theme is somewhat contemporary since it intercepts themes such as biodiversity and species resilience, and ultimately even Blue Economy via the exploitation of intertidal seaweed species. I believe the paper is adequately framed within the wide scope of the Journal, especially in a special issue about Advances in Research of Algae, Cyanobacteria, and Phytoplankton. It should appeal for researchers working in the fields of Marine Biochemistry, mainly those with interest in the study of seaweed intertidal species/ecosystems. The methodology used is well explained and seems sound, bibliography seems up-to-date, and this paper could represent a valuable resource to researchers searching to pursue this interesting theme of how changing environmental conditions affect seaweed composition and metabolism, and maybe even how they modulate output in terms of putative compounds of interest . There are some issues with the paper, but I believe they are manageable and solvable; therefore, I find the paper interesting enough and suitable for publication after the revisions proposed.
Main comments:
My main concern arises from the mentioned “intimate association” with Stigmidium ascophylli. Not being an expert in these types of (symbiotic) associations, I wonder how tight this association is, and ultimately if we may have fungus contamination while analyzing seaweed biomass, namely at the metabolome level. Can we guarantee there is no contamination? If so, were there any cleaning/washing procedures that assure this? Moreover, may the association between fungus and seaweed be favored in any of the tidal conditions, not only increasing the risk of contamination, but also increasing the protective effects hinted in the manuscript? I think this issue could be discussed and clarified.
English in the manuscript is generally adequate but could be improved. There are some problems at times, sometimes with sentences that are not that clear. I included some of those instances in my suggestions.
Minor comments:
Line 52: Start with “An interesting...”
Line 80: “…which are to the moment practically unstudied.” Does not sound right. Consider rewrite, maybe something like “which remain obscure at the moment.”
Line 137: “Algal” instead of “Agal”.
Figure 5: Maybe you could consider converting relative content percentage in units (mg g-1 DW) for pheophytin a, in order to maintain coherence with the graphs for the other pigments.
Table S1: Is there a reason why Table S1 only presents compound quantities for the HW condition? It would be much more informative to have available results for all condition, especially taking into account that apparently many compounds are not even present in the HW condition.
Line 268: “thus promoted its colonizing a new habitat.” Replace with “thus promoted the colonization of a new habitat.”
Line 270: Start the phrase with “The poikilohydrous nature of...”
Line 306: Start with “The amount...”
Line 367: “affected” instead of “effected”.
Line 371: “in the beginning of the waterless period (HW phase)”. Do you mean "LW phase” instead?
Line 389: “Besides the salicylate precursor, during the underwater period Pelvetia cells accumulated also the metabolically linked precursors of tocopherols, such as tyrosine and homogentisic acid”. There is no record of tyrosine being quantified and changed in the experiments, so you should rephrase this: “Besides the salicylate precursor, during the underwater period Pelvetia cells also accumulated homogentisic acid, a metabolically linked precursor of tocopherols.”
References: Please have a thorough look, since there are some references not standardized as the others (for instance, in references 11 and 80 the year of publication is not in bold).
English in the manuscript is generally adequate but could be improved. There are some problems at times, sometimes with sentences that are not that clear. I included some of those instances in my suggestions.
Author Response
We thank the reviewer for his/her attention to our study and valuable comments, which helped us to improve the paper.
All the changes in the text of the manuscript are made using the “Track Changes” function of MS Word.
Reviewer report:
The authors present an interesting approach to study how changing cyclical environmental conditions affect the composition and metabolism of an intertidal seaweed, in Pelvetia canaliculata. The work as merit, its theme is somewhat contemporary since it intercepts themes such as biodiversity and species resilience, and ultimately even Blue Economy via the exploitation of intertidal seaweed species. I believe the paper is adequately framed within the wide scope of the Journal, especially in a special issue about Advances in Research of Algae, Cyanobacteria, and Phytoplankton. It should appeal for researchers working in the fields of Marine Biochemistry, mainly those with interest in the study of seaweed intertidal species/ecosystems. The methodology used is well explained and seems sound, bibliography seems up-to-date, and this paper could represent a valuable resource to researchers searching to pursue this interesting theme of how changing environmental conditions affect seaweed composition and metabolism, and maybe even how they modulate output in terms of putative compounds of interest. There are some issues with the paper, but I believe they are manageable and solvable; therefore, I find the paper interesting enough and suitable for publication after the revisions proposed.
Main comments:
My main concern arises from the mentioned “intimate association” with Stigmidium ascophylli. Not being an expert in these types of (symbiotic) associations, I wonder how tight this association is, and ultimately if we may have fungus contamination while analyzing seaweed biomass, namely at the metabolome level. Can we guarantee there is no contamination? If so, were there any cleaning/washing procedures that assure this? Moreover, may the association between fungus and seaweed be favored in any of the tidal conditions, not only increasing the risk of contamination, but also increasing the protective effects hinted in the manuscript? I think this issue could be discussed and clarified.
RESPONSE:
Though the Pelvetia-Stigmidium association is not regarded as lichen, it is actually intimate enough: according to literature data, the thin hyphae of Stigmidium penetrate the whole algal thallus forming a network inside (Kingham, Evans, 1986; Konovalova et al., 2012). Thus, it is impossible to “clean” the alga from the fungus, and most probably cells of both organisms occurred in all our samples. However, in this association the algal symbiont is definitely a dominant one, and the total biomass of the fungus is several orders of magnitude lower than that of the alga. That’s why, only the fungal metabolites with extremely high concentrations may interfere the analysis and considerably contribute to the measured values. In our earlier works (Tarakhovskaya et al., 2017; Birkemeyer et al., 2019) we studied the metabolome of the other fucoid alga, Fucus vesiculosus, which does not form mycophycobiosis, and the contents of most of the identified metabolites in this alga were very similar to the values, measured in Pelvetia. Moreover, we could not find in Pelvetia samples any metabolites, specific for fungi and not reported for algae. Thus, we think that most of our data actually refer to the alga, and not to the fungus. Nevertheless, we totally agree with the reviewer, that we should emphasize in the paper, that there might be a “fungal contamination” in the algal samples, so we added an appropriate paragraph to the Discussion section (Lines 277-282). Dividing the alga and the fungus and analyzing them separately in different conditions (e.g., different tidal phases) would be a very interesting study indeed, but currently we don’t know the appropriate method for this.
English in the manuscript is generally adequate but could be improved. There are some problems at times, sometimes with sentences that are not that clear. I included some of those instances in my suggestions.
We thank the reviewer for help with English! We accepted all the suggested correction and hope that now the manuscript looks better.
Minor comments:
Line 52: Start with “An interesting...”
RESPONSE:
Corrected.
Line 80: “…which are to the moment practically unstudied.” Does not sound right. Consider rewrite, maybe something like “which remain obscure at the moment.”
RESPONSE:
Corrected.
Line 137: “Algal” instead of “Agal”.
RESPONSE:
Corrected.
Figure 5: Maybe you could consider converting relative content percentage in units (mg g-1 DW) for pheophytin a, in order to maintain coherence with the graphs for the other pigments.
RESPONSE:
The content of pheophytin is now expressed as mg g-1 DW.
Table S1: Is there a reason why Table S1 only presents compound quantities for the HW condition? It would be much more informative to have available results for all condition, especially taking into account that apparently many compounds are not even present in the HW condition.
RESPONSE:
Initially we presented only the HW data because these results could be most easily compared with the values obtained for the other algae (which are mostly taken from underwater). But we agree with the reviewer that the more information the better, so we added the compound quantities for the other tidal phases.
However, this is not the case, that many compounds were not present in the HW condition – the absence of the content value in the Table S1 means that we have no absolute quantitation data for this compound, so it can be quantified only using arbitrary units. We could obtain the absolute quantitation data only for the compounds for which we had the calibration curves made by the standard addition method (about 30 compounds). For the other metabolites (in particular, for the compounds which can’t be identified unambiguously) only relative quantitation is possible. We explained this in the revised version of Table S1.
Line 268: “thus promoted its colonizing a new habitat.” Replace with “thus promoted the colonization of a new habitat.”
RESPONSE:
Corrected.
Line 270: Start the phrase with “The poikilohydrous nature of...”
RESPONSE:
Corrected.
Line 306: Start with “The amount...”
RESPONSE:
Corrected.
Line 367: “affected” instead of “effected”.
RESPONSE:
Corrected.
Line 371: “in the beginning of the waterless period (HW phase)”. Do you mean "LW phase” instead?
RESPONSE:
Yes, we meant “LW phase”, thank you! Now it is corrected.
Line 389: “Besides the salicylate precursor, during the underwater period Pelvetia cells accumulated also the metabolically linked precursors of tocopherols, such as tyrosine and homogentisic acid”. There is no record of tyrosine being quantified and changed in the experiments, so you should rephrase this: “Besides the salicylate precursor, during the underwater period Pelvetia cells also accumulated homogentisic acid, a metabolically linked precursor of tocopherols.”
RESPONSE:
Tyrosine was quantified in our study and the data are presented at Figure 10. This compound was not included in the heatmap (Figure 8), because that picture was already very large, and we did not want to overload it. Thus, tyrosine as well as alpha, gamma, and delta-tocopherols are presented only at Figure 10.
References: Please have a thorough look, since there are some references not standardized as the others (for instance, in references 11 and 80 the year of publication is not in bold).
RESPONSE:
We double-checked all the references. The references 11 and 80 are not articles, but book chapters – that’s why they are formatted differently (it is in accordance with the journal rules).
Used literature:
- Birkemeyer, C.; Osmolovskaya, N.; Kuchaeva, L.; Tarakhovskaya, E. Distribution of natural ingredients suggests a complex network of metabolic transport between source and sink tissues in the brown alga Fucus vesiculosus. Planta 2019, 249, 377–391.
- Kingham, D.L.; Evans, L.V. The Pelvetia-Mycosphaerella In The Biology of Marine Fungi; Moss, S.T., Ed.; Cambridge University Press: Cambridge, UK, 1986, pp. 177–187.
- Konovalova, O.P.; Bubnova, E.N.; Sidorova, I.I. Biology of Stigmidium ascophylli – fungal symbiont of fucoids in Kandalaksha bay, White Sea. Mikol Fitopatol. 2012, 46, 353–360.
- Tarakhovskaya, E.; Lemesheva, V.; Bilova, T.; Birkemeyer, C. Early embryogenesis of brown alga Fucus vesiculosus is characterized by significant changes in carbon and energy metabolism. Molecules 2017, 22, 1509.
Round 2
Reviewer 1 Report
Authors on adressed on comments and now manuscript on suitable on publication.